# Clinical Characteristics of ICI-Related Pancreatitis and Cholangitis Including Radiographic and Endoscopic Findings

**DOI:** 10.3390/healthcare10050763

**Published:** 2022-04-20

**Authors:** Ryota Nakano, Hideyuki Shiomi, Aoi Fujiwara, Kohei Yoshihara, Ryota Yoshioka, Shoki Kawata, Shogo Ota, Yukihisa Yuri, Tomoyuki Takashima, Nobuhiro Aizawa, Naoto Ikeda, Takashi Nishimura, Hirayuki Enomoto, Hiroko Iijima

**Affiliations:** Division of Gastroenterology and Hepato-Biliary-Pancreatology, Department of Internal Medicine, Hyogo College of Medicine, 1-1 Mukogawa-cho, Nishinomiya 663-8501, Hyogo, Japan; ri-nakano@hyo-med.ac.jp (R.N.); aoi.fujiwara0326@gmail.com (A.F.); ko-yoshihara@hyo-med.ac.jp (K.Y.); ri-yoshioka@hyo-med.ac.jp (R.Y.); sh-kawata@hyo-med.ac.jp (S.K.); sy-oota@hyo-med.ac.jp (S.O.); yu-yukihisa@hyo-med.ac.jp (Y.Y.); tomo0204@hyo-med.ac.jp (T.T.); aizawa-n@hyo-med.ac.jp (N.A.); nikeneko@hyo-med.ac.jp (N.I.); tk-nishimura@hyo-med.ac.jp (T.N.); enomoto@hyo-med.ac.jp (H.E.); hiroko-i@hyo-med.ac.jp (H.I.)

**Keywords:** ICI-related pancreatitis, ICI-related cholangitis, irAEs, EUS, ERCP

## Abstract

The indications for immune checkpoint inhibitors (ICIs) have expanded to include carcinomas of various organs. However, as ICI therapy expands, the management of immune-related adverse events (irAEs) has become a problem. ICI-related pancreatitis and cholangitis are relatively rare irAEs. Although some patients with ICI-related pancreatitis and cholangitis are asymptomatic and do not require treatment, there have been reports of patients who did not respond to immunosuppressive therapy and died. Thus, the pathogenesis of ICI-related pancreatitis and cholangitis should be clarified immediately. Currently, the role of endoscopy in the diagnosis and treatment of inflammatory pancreatic and biliary duct diseases is becoming increasingly important. In this review, we summarize clinical characteristics as well as radiographic and endoscopic findings of ICI-related pancreatitis and cholangitis.

## 1. Introduction

Cancer treatment is changing rapidly. In recent years, immune checkpoint inhibitors (ICIs) that target the immune checkpoints used by cancers to reduce immune activity have been developed as new therapeutic agents for cancer immunotherapy. ICIs include drugs that target programmed cell death-1 (PD-1), programmed cell death ligand 1 (PDL-1), and cytotoxic T-lymphocyte-associated antigen 4 (CTLA-4). The indications for these drugs have been expanded to include many types of cancer, as efficacies have been reported for malignant melanoma and lung, kidney, head and neck, stomach, liver, ovarian, and pancreatic cancers [1,2,3]. Commonly used ICIs include PD-1 inhibitors, such as nivolumab and pembrolizumab; PDL-1 inhibitors, such as atezolizumab and avelumab; and CTLA-4 inhibitors, such as ipilimumab. 

However, with this expansion of ICIs, immune-related adverse events (irAEs) caused by cancer immunotherapy have been reported [4,5,6,7]. This has included serious and fatal events involving problematic diagnosis and management. The exact pathogenesis of irAEs remains unknown, and their incidence is being measured; therefore, details regarding their occurrence remain unclear. What is known is that, compared with conventional chemotherapy-induced cytotoxicity, irAEs take longer to manifest [8,9]. Further, irAEs occur in various organs, leading to cutaneous, gastrointestinal, and endocrine toxicity, making organ-specific responses important. According to a recent review, the incidence of irAEs was reported to be 10–15% for nivolumab or pembrolizumab monotherapy, 20–30% for ipilimumab, and 55% for ipilimumab in combination with nivolumab [10].

In 2020, the National Comprehensive Cancer Network (NCCN) proposed guidelines to manage irAEs and recommended consultation with appropriate organ specialists for severe irAEs [11]. In gastroenterology, colitis and hepatotoxicity are the most common adverse events, with colitis occurring in 30–40% of patients and hepatotoxicity in 3–9% [12,13]. However, cholangitis and pancreatitis are rare irAEs, and their clinical characteristics remain unknown. Imaging, in addition to symptoms and general blood tests, is essential for diagnosing hepatobiliary and pancreatic inflammatory diseases such as cholangitis and pancreatitis. In addition to contrast-enhanced computed tomography (CT) and magnetic resonance imaging (MRI), endoscopic examinations such as endoscopic ultrasonography (EUS) and contrast-enhanced imaging by endoscopic retrograde cholangiopancreatography (ERCP) have been gaining importance in recent years. EUS has become an essential modality for diagnosing pancreatic mass lesions, and ERCP is the preferred treatment for drainage of acute cholangitis. However, the usefulness of and evidence for radiology and endoscopy for the diagnosis of ICI-related pancreatitis and cholangitis have not yet been established, as there have been few reported cases and they have not been sufficiently investigated. Therefore, in this review, we summarize previous reports of ICI-related pancreatitis and ICI-related cholangitis from their diagnosis to management and review their clinical characteristics in addition to radiographic and endoscopic findings.

## 2. Results: ICI-Related Pancreatitis

During ICI therapy, blood amylase and lipase levels are often elevated [14]. The significance of these elevated pancreatic enzyme levels in the absence of symptoms is unclear, and aggressive treatment is often not warranted. Therefore, it is not recommended to discontinue ICI therapy based on elevated pancreatic enzymes alone [4,6,7]. Some cases of acute pancreatitis with a symptomatic elevation of pancreatic enzymes during ICI therapy have been reported [4,5,15], and the NCCN recommends evaluation for pancreatitis, including imaging studies, if the pancreatic enzyme elevations persist [11]. If the onset of acute pancreatitis is confirmed, the NCCN recommends treatment, including rapid infusion and pain management, and consultation with a specialist for moderate or more severe pancreatitis [11]. Nevertheless, because ICI-related pancreatitis is quite rare and there have been few reported cases, its imaging characteristics and appropriate treatment are largely unknown. Although cases of severe pancreatitis are rare, deaths have been reported [16].

### 2.1. Incidence

ICI-related pancreatitis has been reported as a rare irAE [5,17,18,19]. According to previous reports, the incidence of ICI-related pancreatitis has been reported to be approximately 0.3–3.9% [5,14,17,18,19]. Tirumani et al. reported that in 147 melanoma patients treated with ipilimumab, 46 (31%) developed irAEs, and the incidence of irAE pancreatitis was less than 1% [17]. Additionally, Friedman et al. reported that in a retrospective study of 119 patients with melanoma treated with nivolumab and ipilimumab, the incidence of pancreatitis was 1.7%, occurring in 20% of patients with highly elevated amylase and lipase [18]. Moreover, George et al. found that CTLA-4 inhibitors had a significantly higher incidence of pancreatitis than PD-1 inhibitors (3.98% vs. 0.94%, *p* < 0.05) and that combination therapy with CTLA-4 inhibitors and PD-1 inhibitors resulted in a relatively higher incidence of pancreatitis than monotherapy [20]. All reported cases in which radiographic or endoscopic images of ICI-related pancreatitis were available are shown in Table 1. The onset of pancreatitis varied from 20 days to more than one year after the start of ICI therapy, depending on the case.

### 2.2. Diagnosis

#### 2.2.1. Clinical Symptoms

Similar to the clinical manifestations of acute pancreatitis in general [28], symptoms such as abdominal pain, back pain, nausea and vomiting, diarrhea, and fever have been reported in ICI-related pancreatitis [29]. Epigastric pain was the most frequent symptom in 39% of patients with ICI-related pancreatitis, and many patients had multiple symptoms [30]. However, a certain number of patients were also observed to have pancreatitis noted on imaging but were completely asymptomatic [30].

#### 2.2.2. Blood Examination

Elevated blood pancreatic enzymes are most frequently observed in ICI-related pancreatitis [30]. In patients undergoing treatment with anti-CTLA-4 antibodies and anti- PD-1 antibodies, grades 3–4 of the Common Terminology Criteria for Adverse Events (CTCAE) and elevated levels of serum amylase and lipase were reported in 1–8% [1,18]. Blood lipase levels are considered useful for diagnosing acute pancreatitis, as they have better sensitivity and specificity than amylase levels [31]. Friedman et al. reported that ICI-related pancreatitis occurred in 20% of patients with elevated amylase and 6.7% of patients with elevated lipase levels with a CTCAE grade ≥3 [18]. Abu-Sbeih et al. also reported that among patients with ICI-related pancreatitis, those with clinical symptoms had significantly higher blood lipase levels than asymptomatic patients (*p* = 0.032) [30]. However, many cases of clinically undiagnosed pancreatitis have been observed, even in patients with lipase levels above grade 3. These data indicate that in most cases, lipase increase related to anti-PD1 or anti-PD-L1 is not associated with a significant clinical event. [14,30]. These results suggest that elevated pancreatic enzymes alone are not the basis for the development of ICI-related pancreatitis and that imaging evaluation is important for diagnosis. Autoimmune antibodies, including immunoglobulin G4 (IgG4), have been reported to be within normal ranges [21,22].

#### 2.2.3. Radiology Images

The imaging features of ICI-related pancreatitis have been reported to include two patterns: one resembling acute interstitial pancreatitis and the other resembling autoimmune pancreatitis (Table 1). The imaging features of ICI-related pancreatitis, which resemble those of acute interstitial pancreatitis, include an enlarged pancreas on contrast-enhanced CT, poor contrast, and increased lipid concentrations [30]. Hofmann et al. reported that abdominal CT consistently revealed reduced lobulation, tissue swelling, and reduced tissue contrast enhancement in the pancreatic body and tail [5]. In contrast, Das et al. reported that diffuse (*n* = 14) or localized (*n* = 11) pancreatic enlargement was noted as an imaging finding in ICI-related pancreatitis and that a pattern consistent with autoimmune pancreatitis was present in 4/25 (16%) patients, while no patients developed necrotizing pancreatitis or pancreatic pseudocysts [23]. Regarding MRI, a few characteristic findings of ICI-related pancreatitis have been reported. ICI-associated pancreatitis with imaging findings similar to those of focal type 2 autoimmune pancreatitis has been noted on MRI, including signal restriction on diffusion-weighted images and contrast enhancement in the late phase of gadolinium contrast [22,24] (Table 1). Capurso et al. reported that the pancreatic head showed signal limitation on diffusion-weighted images and stenosis of the main pancreatic duct at the pancreatic head [24]. It has also been reported in several studies that ICI-related pancreatitis has a strong uptake of 18F-fluorodeoxyglucose positron emission tomography combined with CT (18F-FDG-PET/CT) [22,25,26] (Table 1). During ICI therapy, FDG-PET is often performed to evaluate the effect of treatment on the primary tumor, and asymptomatic cases of ICI-related pancreatitis incidentally diagnosed using FDG-PET have also been reported [22,25]. Radiology imaging reports of ICI-related pancreatitis are very limited. However, based on the reported cases of these imaging features, ICI-related pancreatitis can be classified into two major patterns: cases with imaging findings similar to acute interstitial pancreatitis and cases with imaging findings similar to type 2 autoimmune pancreatitis. During ICI treatment, when characteristic imaging findings of acute interstitial pancreatitis or type 2 autoimmune pancreatitis are observed, ICI-related pancreatitis should be considered a differential diagnosis (Table 1).

### 2.3. Endoscopic Findings

#### 2.3.1. EUS

EUS has been established as a standard diagnostic imaging modality for pancreatic tumors and inflammatory pancreatic diseases and has become an indispensable modality for the detailed observation of pancreatic diseases. Some EUS images of ICI-related pancreatitis have been reported as similar to those characteristic of autoimmune pancreatitis [21,24,27,32,33] (Table 1 and Figure 1). Tanaka et al. reported nivolumab-related pancreatitis with imaging findings similar to diffuse-type autoimmune pancreatitis, and EUS showed diffuse hypoechoic enlargement of the pancreas with a hypoechoic, patchy, and heterogeneous parenchyma [27]. Ofuji et al. reported that in pembrolizumab-related pancreatitis, EUS showed an enlarged pancreas with hypoechogenicity and scattered hyperechoic spots [21]. Capurso et al. reported EUS findings mimicking those of focal-type autoimmune pancreatitis. Their report showed a 2 cm hypoechoic solid lesion of the pancreatic neck, stiffness on elastography, and low vascularity after the administration of a contrast agent in EUS images [24]. Some cases of ICI-related pancreatitis have been reported to be similar to the CT and EUS images of autoimmune pancreatitis [23]. In the diagnosis of autoimmune pancreatitis, numerous characteristic EUS findings have been reported for both diffuse and focal types. The diffuse type is characterized by pancreatic enlargement with diffuse hypoechoic images having a sausage-like appearance. Conversely, a capsule-like hypoechoic zone at the edge of the mass or spot-like hyperechoic findings in the pancreas may be observed in the focal type. These EUS findings are assumed to reflect a high degree of inflammatory cell infiltration. Distinguishing focal type pancreatitis from pancreatic cancer is particularly difficult because both appear as hypoechoic masses. Compared to pancreatic cancer, autoimmune pancreatitis may appear as multiple masses, and the duct penetration sign, which shows the pancreatic duct within the mass, is considered useful for diagnosing autoimmune pancreatitis.

#### 2.3.2. ERCP

Generally, ERCP is not performed to diagnose acute pancreatitis, except in cases of gallstone pancreatitis. There are few reports of ERCP performed for ICI-related pancreatitis, and imaging findings of such ERCP procedures are almost unknown. Tanaka et al. reported the endoscopic diagnosis of nivolumab-related pancreatitis using ERCP. According to these, pancreatic duct findings by ERCP showed a relatively long narrowing with a skip in the main pancreatic duct, which was very similar to the findings in autoimmune pancreatitis [27]. Irregular narrowing of the main pancreatic duct is considered a characteristic of autoimmune pancreatitis on pancreatic ductal angiography with ERCP [34]. Irregular narrowing of the main pancreatic duct is defined as a diffuse or localized finding of a relatively long series of narrowed and irregular pancreatic ducts rather than the localized obstruction or stenosis characteristic of pancreatic ductal carcinoma. In addition, mild dilatation of the pancreatic duct of the pancreatic tail is also characteristic of autoimmune pancreatitis compared to pancreatic ductal carcinoma [35].

#### 2.3.3. Histopathological Findings

EUS-guided fine-needle aspiration (EUS-FNA) has been shown to have a high diagnostic capability for pancreatic masses and is currently performed as an essential test for their differential diagnosis. In the absence of surgical resection, EUS-FNA is the only modality used for histological evaluation of pancreatic disease. In recent years, with the improvement in puncture needle technology, the number of specimens that can be collected by EUS-FNA has been increasing, making it possible to perform a thorough pathological evaluation. Pathological studies using EUS-FNA have reported several cases of ICI-related pancreatitis [24,32,33,36]. According to the pathological findings of autopsy cases of ICI-related pancreatitis, along with the pathological findings of pembrolizumab necrotizing acute pancreatitis, significantly more CD8+ T cells than CD4+ T cells were detected in the remaining pancreatic parenchyma [16]. A report of EUS-FNA for pembrolizumab-related pancreatitis also showed infiltration of T-lymphocytes, with a predominance of CD8+ cells over CD4+ cells [36]. Ofuji et al. reported that EUS-FNA was performed for ICI-related pancreatitis with a 25-G needle puncture, and cytological examination showed infiltration of inflammatory cells composed mainly of neutrophils, which is a presentation similar to that seen in type 2 autoimmune pancreatitis [21]. Song et al. reported that when EUS-FNA was performed for nivolumab-related pancreatitis, EUS images showed mass-forming pancreatitis, with lymphocytes infiltrating the surrounding pancreatic duct and fibrosis in the stroma and no plasma cells. These authors also highlighted the similarities between nivolumab-related and type 2 autoimmune pancreatitis [33].

### 2.4. Management

Steroid therapy is generally used to treat irAEs that develop during ICI therapy. The NCCN guidelines recommend that ICI therapy continuation should be considered when there is no evidence of pancreatitis and only elevated amylase and lipase levels are observed [11]. For moderate pancreatitis, the NCCN recommends interruption of ICI therapy and initiation of methylprednisolone or prednisolone at 0.5–1.0 mg/kg/day, and for severe pancreatitis, complete discontinuation of ICI therapy and steroids at 1–2 mg/kg/day. However, Abu-Sbeih et al. reported that steroid therapy for ICI-associated pancreatitis did not affect short-term improvement in lipase levels or shorten hospital stay, making the efficacy of steroid therapy controversial [30]. In nearly all reported cases, ICI-related pancreatitis showed an eventual improvement by interruption of ICI therapy or steroid therapy. A severe acute case of ICI-related pancreatitis was diagnosed after the administration of nivolumab followed by pazopanib. According to the report, methylprednisolone was administered, and conservative treatment for acute pancreatitis was initiated, but the patient’s general condition rapidly worsened, resulting in a fatality [16].

## 3. Results: ICI-Related Cholangitis

Liver injury is a relatively common IrAE. The incidence of ICI-related hepatotoxicity is estimated to be 3–9% for ipilimumab and 0.7–1.8% for PD-1/PD-L1 inhibitors [13], with its incidence reported to increase further when multiple drugs are used [13,37]. Among ICI-related hepatobiliary disorders, cholangitis is rarely reported, and its pathogenesis is unknown. However, in recent years, there have been scattered reports of ICI-related cholangitis that develops as sclerosing cholangitis. Currently, dozens of cases of ICI-related sclerosing cholangitis have been reported, often presenting images similar to those of IgG4-related sclerosing cholangitis and primary sclerosing cholangitis (PSC). Based on previous reports, we analyzed the diagnosis and management of ICI-related cholangitis in detail.

### 3.1. Incidence

Although irAE hepatotoxicity is a relatively common adverse event, cholangitis associated with ICIs is very rare, and its incidence is unknown. In 2017, Gelsomino et al. first reported a case of cholestatic liver injury that occurred 62 days after nivolumab treatment in non-small cell lung cancer [38]. In this report, a percutaneous liver biopsy was performed, which showed images of drug-induced liver damage with findings of cholangitis. Kawashima et al. reported that of 91 patients treated with nivolumab for non-small cell lung cancer, 3.3% (three patients) developed ICI-related cholangitis [39]. All reported cases in which radiographic or endoscopic images of ICI-related cholangitis were available are shown in Table 2. Most previous reports of ICI-associated cholangitis have been related to PD-1 antibodies (excluding one case with avelumab). Although it is unclear why PD-1 inhibitors alone cause ICI-related cholangitis, they may be related to mutations in the gene that encodes a specific immune checkpoint protein. The onset of ICI-associated cholangitis varies among cases. Some patients developed cholangitis within a very short period, such as after the first course of nivolumab, while others developed cholangitis after 11 months of nivolumab.

### 3.2. Diagnosis

#### 3.2.1. Clinical Symptoms

In cases of acute cholangitis, the most common clinical symptoms are intermittent fever, right upper quadrant pain, and jaundice [53]. As with obstructive cholangitis, ICI-related cholangitis often leads to clinical symptoms such as fever, abdominal pain, general malaise, and vomiting [39,40,41,42,43,44]. In contrast, some cases are asymptomatic and do not necessarily show clinical symptoms [54].

#### 3.2.2. Blood Examination

In blood tests, biliary system enzymes are often reported to be predominantly elevated compared to liver enzymes. In a report by Kawakami et al., nivolumab-related cholangitis was characterized by a dominant increase in the biliary tract enzymes alkaline phosphatase (ALP) and gamma-glutamyl transpeptidase (r-GTP) relative to the hepatic enzymes aspartate aminotransferase (AST) and alanine aminotransferase (ALT) [39]. Summarizing the previously reported cases, the peak values of ALP and r-GTP were 1543–3096 U/L and 252–1774 U/L, respectively, which were predominantly elevated over the peak values of AST and ALT of 69–313 U/L and 68–296 U/L, respectively [39,40,42,43,45,46,54]. Regarding immunological findings in ICI-associated cholangitis, Kawakami et al. reported normal or decreased levels of serum immunological markers such as antinuclear antibody, anti-mitochondrial antibody, smooth muscle antibody, and IgG4 [39].

#### 3.2.3. Radiology Images

Kawakami et al. reported that nivolumab-related cholangitis is characterized by the following findings: (1) localized extrahepatic bile duct dilation without obstruction; (2) diffuse hypertrophy of the extrahepatic bile duct wall; (3) dominant increase in the biliary tract enzymes ALP and r-GTP relative to the hepatic enzymes AST and ALT; (4) normal or reduced levels of the serum immunological markers antinuclear antibody, antimitochondrial antibody, smooth muscle antibody, and immunoglobulin G4; (5) pathological findings of biliary tract CD8-positive T-cell infiltration from a liver biopsy; and (6) a moderate to poor response to steroid therapy [39]. The imaging features of (1) and (2) are seen on CT or MRI, which mainly appear based on changes in the “extrahepatic” bile ducts. Several other studies have reported the appearance of diffuse hypertrophy of the extrahepatic bile duct walls on contrast-enhanced CT [39,41,43,46,47,48,49] (Table 2). Recently, there have also been reports of ICI-related cholangitis with imaging changes in “intrahepatic” bile ducts [41,42,43,44,45,46,48,50,51,55,56] (Table 2). Therefore, it is necessary to consider the presence of ICI-associated cholangitis, which causes inflammation not only in the extrahepatic bile ducts but also in the intrahepatic bile ducts (Figure 2). In a study by Koya et al., magnetic resonance cholangiopancreatography (MRCP) revealed an irregularly narrowed intrahepatic bile duct and dilated peripheral bile ducts [41]. McClure et al. highlighted the beading sign of intrahepatic bile ducts on MRCP [42], and Hirasawa et al. reported that MRCP showed beaded stenosis of the intrahepatic bile ducts in imaging of ICI-related cholangitis in patients refractory to immunosuppressive therapy [52]. Considering the imaging findings of ICI-related cholangitis seen on CT and MRI, the beading sign of the distal bile duct and hypertrophy of the extrahepatic bile duct wall are similar to those observed in IgG4-related cholangitis. On the contrary, multiple stenoses, such as a beading sign of the intrahepatic bile ducts (or both intrahepatic and extrahepatic bile ducts), can cause PSC (Table 2). To summarize the reported radiographic features of ICI-related cholangitis, the image types of ICI-related cholangitis can be classified into two types: patterns similar to PSC and patterns similar to IgG4SC. In some reports, contrast-enhanced CT showed gallbladder wall thickening and edema along with bile duct wall thickening in the common bile duct, which were suspected to be complications of cholecystitis [40,43,44,48].

### 3.3. Endoscopic Findings

#### 3.3.1. EUS

EUS plays an important role in diagnosing sclerosing cholangitis, including IgG 4-related cholangitis and PSC. From the evaluation of bile duct stenosis and dilatation, as well as the finding of wall thickening, it is possible to obtain a considerable amount of information about the main site of inflammation, the presence or absence of obstruction, and its differentiation from neoplastic lesions. There are some reports of ICI-related cholangitis in which EUS was performed [39,41,52] (Table 2). In the cases reported by Kawakami et al., EUS was performed for three cases of ICI-related cholangitis, and diffuse enlargement of extrahepatic bile ducts was found in all cases [39]. Hirasawa et al. reported that EUS for nivolumab-related cholangitis revealed extrahepatic bile duct dilation and gallbladder wall thickening [52]. Ogawa et al. reported that EUS revealed significant caliber fluctuations and irregular wall thickness in the intrahepatic and extrahepatic bile ducts [46].

#### 3.3.2. ERCP

ERCP has been performed for ICI-related cholangitis [39,40,41,47,48,52]. Some reports describe irregularities in the bile duct wall of the extrahepatic and intrahepatic bile ducts on CT or MRI bile duct imaging [39,41,47,49,51] (Table 2). Koya et al. reported that during ERCP, there were findings of spontaneous bleeding from the ampulla of Vater, with contrast images showing irregularities in the walls of the intrahepatic and extrahepatic bile ducts as well as irregular narrowing of the intrahepatic bile ducts [41]. We suggest that ICI-related cholangitis presents imaging findings similar to those of PSC and IgG4-SC. In some cases, it is difficult to differentiate PSC from IgG4-SC on imaging, but when band-like stricture, beaded appearance, and diverticulum-like outpouching findings by MRCP or ERCP are seen, they are reported to be characteristic of PSC. In contrast, segmental or long stricture with prestenotic dilatation is reported to be significantly more common in IgG4-SC. Although ERCP is an invasive endoscopic procedure, for long bile duct stricture, it is important to differentiate benign inflammatory strictures, such as ICI-associated cholangitis, from malignant biliary strictures, which require tissue sampling such as bile duct biopsy or brush cytology by ERCP.

In recent years, the usefulness of peroral cholangioscopy for the diagnosis and treatment of biliary tract diseases has been widely reported. Endoscopic findings of irAEs using peroral cholangioscopy have also been reported, although in only a few cases [41,47]. Koya et al. reported peroral cholangioscopy endoscopic findings for irAE of multiple scarred lesions of the extrahepatic bile ducts, which showed a hemorrhagic state and a narrowing of second-order biliary branches [41]. Onoyama et al. performed peroral cholangiography in three cases of pembrolizumab-related sclerosing cholangitis and reported findings of areas of band-like narrowing of the wall of the biliary tract and diverticulum-like outpouching [47]. Kuraoka et al. reported nivolumab-associated cholangitis that caused severe inflammatory changes, with peroral cholangioscopy revealing erosions with black spots, inflammation, and yellow plaques [49].

Endoscopic biliary drainage is recommended for patients with moderate or severe disease [53], but its efficacy in ICI-associated cholangitis is unknown. Although there have been several reports of endoscopic drainage for ICI-associated cholangitis, many reported that it did not contribute to the radical cure of cholangitis, suggesting that bile duct drainage may not be useful in the treatment of ICI-associated cholangitis [39,40,48,52]. In the case of Kawakami et al., ERCP showed gentle beak-like stenosis without distal bile duct obstruction. Endoscopic bile duct drainage with plastic stenting was performed for ICI-related infectious cholangitis; however, there was no improvement after the procedure [39]. Kawashima et al. reported that endoscopic nasal drainage improved abdominal symptoms but did not decrease biliary enzyme elevation [40].

#### 3.3.3. Histopathological Findings

Some reports detail the pathological analysis of liver biopsies in ICI-related cholangitis. In most reports, T-cells infiltrated the bile ducts, with CD8+ T cells predominating over CD4+ T cells [39,43,45,50,52,54,57]. Several cases of ERCP with repeat bile duct biopsy have been reported, including infiltration of inflammatory CD4+ and CD8+ T cells into the stroma [40,41,46,52]. Zen et al. investigated the pathological changes of pembrolizumab- and atezolizumab-induced hepatobiliary reactions and pointed out that the ratio of CD8+/CD4+ cells is significantly higher than that in autoimmune or drug-induced hepatitis. They reported one case showing diffuse severe bile duct inflammation similar to primary sclerosing cholangitis; however, the histopathological characteristics of the bile duct biopsy specimen were similar to those of IgG4-related cholangitis, as the inflammation was mainly located inside the duct wall and the epithelial layer was relatively preserved [57]. It has been pointed out that ICI-related cholangitis, which shows diffuse inflammation similar to primary sclerosing cholangitis, may not respond well to steroid therapy [57]. Histopathological evaluation by liver biopsy or endoscopic bile duct biopsy may be useful in determining the response to treatment. Further histopathological evaluation by liver biopsy or endoscopic bile duct biopsy by ERCP is required to determine the response to treatment for cholangitis.

### 3.4. Management

Steroids and combined immunosuppressive therapies are generally recommended to treat irAEs, but there is no consensus on the treatment of ICI-related cholangitis owing to the lack of accumulated cases. In many cases, oral prednisolone and ursodeoxycholic acid (UDCA) are administered, but their therapeutic effects vary. There are reports of improvement in hepatic damage with drug discontinuation alone [42,46], as well as immediate or gradual recovery of hepatobiliary enzyme levels after oral administration of prednisolone [42,43,50,54]. The dosage of oral steroids is often 0.5–1.0 mg/kg/day, but there are some reported cases where the dosage was increased to 2.0–3.0 mg/kg/day or steroid pulses were administered. Thus, there is no consensus on the administration of steroid therapy. In a report by Kawakami et al., prednisolone 0.5 mg/kg/day was administered for ICI-associated cholangitis without infection and tapered over 1–2 weeks, after which marked improvement in hepatobiliary enzymes was observed, eventually allowing resumption of ICI therapy [39]. In the case of Kawashima et al., high-dose prednisone (1 mg/kg/day) was started, and the maximum prednisone dose was increased to 2 mg/kg/day, with an improvement in imaging on day 49 after the start of prednisone administration [40]. On the contrary, there have been several reports of patients with a poor response to steroids and UDCA, showing no improvement despite administering a combination immunosuppressive therapy. Amawi et al. reported a case of prednisolone-resistant pembrolizumab-associated cholangitis in which intrahepatic bile duct stenosis and dilatation worsened over time, with uncontrolled jaundice [48]. Recently, four cases of nivolumab-induced sclerosing cholangitis were reported in which combination immunosuppressive therapy (prednisolone, mycophenolate mofetil, and tacrolimus) was not effective; moreover, a mortality case due to liver failure was reported [45,49,52,56]. As shown in Table 2, 10 cases with bead appearance/multiple stenoses in the intrahepatic bile ducts on MRCP/ERCP were reported; 60% (6/10) of the cases showed a low response to treatment, and all fatal cases showed intrahepatic bile duct imaging findings. Thus, MRCP/ERCP imaging findings may play an important role in response to immunosuppressive therapy.

## 4. Conclusions

ICI-related pancreatitis and cholangitis are both infrequent irAEs. However, in the past few years, the number of reported cases has increased, and the pathogenesis of these diseases has gradually become clearer. Nonetheless, evidence from only a small number of case reports is available, and many aspects of diagnosis and treatment remain unclear. Some fatal cases have not responded to immunosuppressive therapy, and further accumulation of case reports and studies to clarify the pathogenesis of the disease is still needed. Although the clinical features of ICI-related pancreatitis and cholangitis are largely unknown, through this review, the characteristic radiographic and endoscopic findings of ICI-related pancreatitis and cholangitis have been identified. Imaging studies of ICI-related pancreatitis suggest two imaging patterns: acute interstitial pancreatitis and autoimmune pancreatitis. Furthermore, imaging studies of ICI-related cholangitis suggested two imaging patterns, PSC and IgG4-SC. Histopathological features of both ICI-related pancreatitis and ICI-related cholangitis were reported to show infiltration of CD8+ T cells, suggesting that histological evaluation by EUS-FNA plays a very important role in the imaging pattern of autoimmune pancreatitis. Although the efficacy of bile duct drainage for ICI-associated cholangitis is not clear, it is suggested that bile duct biopsy by ERCP and endoscopic diagnosis by oral cholangioscopy may be useful in the diagnosis of ICI-associated cholangitis. In this review, we summarized reported cases of ICI-related pancreatitis and cholangitis and reviewed their clinical characteristics as well as radiographic and endoscopic features.

## Figures and Tables

**Figure 1 healthcare-10-00763-f001:**
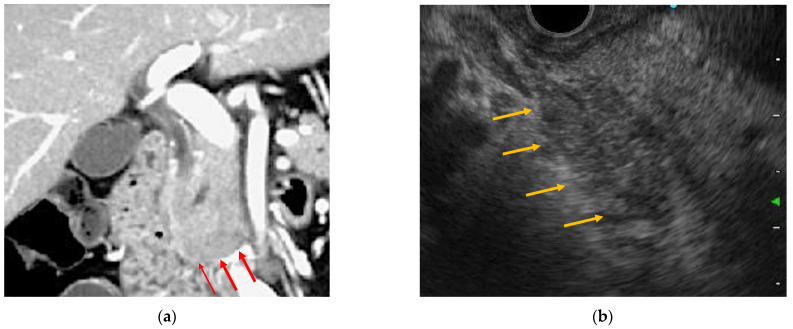
Endoscopic ultrasonography and contrast-enhanced computed tomography (CT) images of immune checkpoint inhibitor (ICI)-related pancreatitis. (**a**) Focal pancreatic enlargement at the pancreatic head with a mass-like lesion and poor contrast enhancement (arrowhead). (**b**) Focal hypoechoic mass-like findings with internal hyperechoic spots from the pancreatic head to the pancreatic uncinate process (arrowhead).

**Figure 2 healthcare-10-00763-f002:**
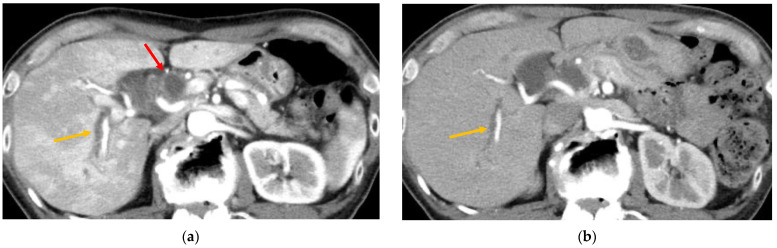
Contrast-enhanced computed tomography (CT) findings of immune checkpoint inhibitor (ICI)-related cholangitis. (**a**) In the arterial phase, the liver parenchyma is irregularly enhanced. Wall thickening of the extrahepatic bile duct (red, arrowhead) and dilatation of the intrahepatic bile ducts (orange, arrowhead) were also observed. (**b**) After starting steroid therapy, the irregularly enhanced hepatic parenchyma showed a marked improvement. Intrahepatic bile duct dilatation remained but gradually improved (orange, arrowhead).

**Table 1 healthcare-10-00763-t001:** Radiographic and endoscopic images of ICI-related pancreatitis.

No.	Ref.	Sex	Age	ICI	CT Findings	MRI Findings	EUS Findings	ERCP Findings	Imaging Type
1	Ofuji et al. [21]	M	82	pembrolizumab	diffuse enlargement	diffuse restricted diffusion diffuse enlargement narrowing of the MPD	hypoechoic enlargement hyperechoic spots	NA	autoimmune pancreatitis
2	Dehghani et al. [22]	M	63	nivolumab	focal enlargement fat stranding	focal restricted diffusion late enhancement	NA	NA	autoimmune pancreatitis
3	Das et al. [23]	M	47	nivolumab	diffuse enlargement diffuse fat stranding	NA	NA	NA	acute interstitial pancreatitis
4	Das et al. [23]	F	70	nivolumab	focal enlargement subtle fat stranding	NA	NA	NA	acute interstitial pancreatitis
5	Das et al. [23]	F	50	pembrolizumab	NA	focal enlargement abrupt cut-off of the CBD	NA	NA	autoimmune pancreatitis
6	Das et al. [23]	F	64	nivolumab	diffuse enlargement heterogenous enhancement fat stranding	NA	NA	NA	acute interstitial pancreatitis
7	Das et al. [23]	F	56	ipilimumab nivolumab	NA	NA	NA	NA	autoimmune pancreatitis
8	Capurso et al. [24]	F	76	pembrolizumab	MPD dilation	MPD dilation focal restricted diffusion	hypoechoic solid lesion stiff at elastography stenosis of the MPD	NA	autoimmune pancreatitis
9	Saito et al. [25]	M	72	nivolumab	diffuse enlargement	NA	NA	NA	acute interstitial pancreatitis
10	Kakuwa et al. [26]	M	70	pembrolizumab	mild diffuse enlargement MPD dilation	NA	NA	NA	autoimmune pancreatitis
11	Tanaka et al. [27]	F	70	nivolumab	NA	diffuse enlargement focal restricted diffusion	diffuse hypoechoic enlargement	skipped narrowing of the MPD	autoimmune pancreatitis

Clinical characteristics and imaging findings of ICI-related pancreatitis cases in which radiographic and endoscopic images are available. Abbreviations: Ref., reference; ICI, immune checkpoint inhibitor; CT, computed tomography; MRI, magnetic resonance imaging; EUS, endoscopic ultrasonography; ERCP, endoscopic retrograde cholangiopancreatography; PET-CT, positron emission tomography combined with computed tomography; MPD, main pancreatic duct; CBD, common bile duct; NA, not available.

**Table 2 healthcare-10-00763-t002:** Radiographic and endoscopic images of ICI-related cholangitis.

No.	Ref.	Sex	Age	ICI	CT Findings	MRI Findings	EUS Findings	ERCP Findings	Imaging Type
1	Kawakami et al. [39]	M	64	Nivolumab	E; dilation, hypertrophy I; NA G; hypertrophy	NA	E; dilation, diffuse hypertrophy I; normal G; hypertrophy	NA	IgG4-SC
2	Kawakami et al. [39]	F	73	Nivolumab	E; dilation I; NA G; hypertrophy	E; dilation I; NA G; NA	E, diffuse hypertrophy I; normal G; NA	E; dilation, distal stenosis I; normal G; NA	IgG4-SC
3	Kawakami et al. [39]	F	82	Nivolumab	E; dilation I; NA G; NA	E; dilation I; NA G; NA	E; dilation, diffuse hypertrophy I; normal G; NA	E; dilation, distal stenosis I; normal G; NA	IgG4-SC
4	Kashima et al. [40]	M	63	Nivolumab	E; dilation, beaking stenosis I; NA G; hypertrophy	E; dilation, beaking I; normal G; normal	NA	NA	IgG4-SC
5	Koya et al. [41]	M	66	Pembrolizumab	E; hypertrophy I; NA G; NA	E; normal I; multiple irregular narrowing G; NA	E; diffuse hypertrophy I; NA G; NA	E; irregularity of the bile duct I; multiple irregular narrowing G; NA	PSC
6	McClure et al. [42]	M	79	Nivolumab	NA	E; NA I; beaded appearance G; NA	NA	NA	PSC
7	Sato et al. [43]	M	69	Pembrolizumab	E; diffuse hypertrophy I; dilation G; hypertrophy	E; normal I; multiple irregular narrowing G; normal	NA	NA	PSC and IgG-SC
8	Cho et al. [44]	M	69	Avelumab	E; dilation, hypertrophy I; normal G; hypertrophy	NA	NA	NA	IgG4-SC
9	Yoshikawa et al. [45]	M	75	Nivolumab	NA	E; diffuse dilatation I; multifocal stenosis G; normal	NA	NA	PSC
10	Ogawa et al. [46]	M	73	Pembrolizumab	E; hypertrophy I; dilation G; hypertrophy	NA	E; irregular hypertrophy I; irregular hypertrophy G; NA	E; irregularity of the bile duct I; multiple irregular narrowing G; NA	PSC
11	Onoyama et al. [47]	M	63	Pembrolizumab	E; irregular hypertrophy I; NA G; NA	NA	E; irregular hypertrophy I; NA G; NA	E; irregularity of the bile duct I; normal G; NA	IgG4-SC
12	Tahboub et al. [48]	M	67	Pembrolizumab	E; diffuse hypertrophy I; dilation G; hypertrophy	E; normal I; multiple irregular narrowing G; normal	NA	NA	PSC
13	Kuraoka et al. [49]	M	69	Nivolumab	E; hypertrophy I; NA G; NA	NA	E; diffuse hypertrophy I; NA G; NA	E; irregularity of the bile duct I; multiple irregular narrowing G; NA	PSC and IgG-SC
14	Hamoir et al. [50]	M	71	Nivolumab	E; normal I; normal G; normal	E; normal I; multiple stenosis G; NA	NA	NA	PSC
15	Kono et al. [51]	F	69	Nivolumab	E; hypertrophy I; NA G; hypertrophy	E; NA I; dilation G; hypertrophy	E; NA I; dilation G; hypertrophy	E; stenosis I; multiple irregular narrowing G; NA	PSC and IgG-SC
16	Hirasawa et al. [52]	M	64	Nivolumab	E; diffuse hypertrophy I; dilation G; hypertrophy	E; hypertrophy I; beaded appearance G; normal	E; dilation I; NA G; hypertrophy	E; dilation I; NA G; NA	PSC and IgG-SC

Clinical characteristics and imaging findings of ICI-related cholangitis cases in which radiographic and endoscopic images is available. Abbreviations: Ref., reference; ICI, immune checkpoint inhibitor; CT, computed tomography; MRI, magnetic resonance imaging; EUS, endoscopic ultrasonography; ERCP, endoscopic retrograde cholangiopancreatography; NSCLC, non-small cell lung cancer; E, extrahepatic bile duct; I, intrahepatic bile duct; G, gallbladder; PSC, primary sclerosing cholangitis; IgG4-SC, immunoglobulin G4-related sclerosing cholangitis; NA, not available.

## Data Availability

Not applicable.

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
