# Peer review of "Clinical Characteristics of ICI-Related Pancreatitis and Cholangitis Including Radiographic and Endoscopic Findings"

_healthcare, 2022, doi:10.3390/healthcare10050763_

Round 1

Reviewer 1 Report

Submitted review "Role of endoscopic diagnosis in immune checkpoint inhibitor-related pancreatitis and cholangitis" is an interesting work in the field of immune-related adverse reactions and their diagnostics, however it is in need of substantial editing for grammatical errors prior to reconsideration for publishing. Overall, the article gives a very good, detailed overview of current literature, is clearly organized into sections and logically planned.

The presented paper describes broadly the role of endoscopic diagnosis in immune checkpoint inhibitor-2 related pancreatitis and cholangitis. To begin with, the authors touch on an issue which has not been widely described. As they state themselves in the introduction, rapid development of new cancer treatments presents us also with new challenges linked to adverse events during cancer therapy, however, ICI-induced cholangitis and pancreatitis are not diagnosed differently than these of other aetiology, so while the topic may be novel, the results do not provide a significant advance in current knowledge.

The manuscript is divided into the introduction, ICI-related pancreatitis and cholangitis sections, followed by a conclusion. Both of the main sections are further broken down into subsections explaining incidence, non-endoscopic diagnostic modalities, endoscopy and treatment. While this division is acceptable and allows the reader to follow the authors easily, I feel that too little stress is on the endoscopy itself and too much on describing both diseases in general.

Both tables do not appear clear to me, especially table 1 would benefit from a redesign to fit into one page. What does “imaging type” mean? And what is “outcome” related to, drug discontinuation?

Table 2 – I suggest adding separate rows for E, I and G findings.

This table shows that practically all EUS and ERCP findings were also seen in imaging studies and as EUS and ERCP are invasive techniques, this also makes one doubt the role of endoscopy as a diagnostic modality.

Overall, I would suggest either to rewrite the article to put more emphasis on endoscopy itself and consider changing the title to e.g. “endoscopic presentation of ICI-related pancreatitis and cholangitis”. Endoscopic examinations are rarely meant as first-line diagnostic modalities in cholangitis and pancreatitis, they are preferred when a therapeutic intervention is planned, or in the event of complications.

Finally, as I mentioned in the previous version of my review, the whole manuscript requires major English revision, preferably by an editor with medical language expertise.

Author Response

Thank you very much for your constructive and helpful comments to improve the impact of our manuscript . According to your comments and suggestions, we have modified our manuscript, please see the attachment.

Reviewer 2 Report

The study is aimed to review the role of endoscopic diagnosis in immune checkpoint inhibitor-related pancreatitis and cholangitis.  The title is “Role of endoscopic diagnosis in immune checkpoint inhibitor-related pancreatitis and cholangitis”.

  1. This is a review article.
  2. Several factors influence the endoscopic diagnosis. Please discuss these.
  3. What is the new knowledge of the report?
  4. Please recommend to the readers “How to apply this knowledge?”.

Author Response

(The authors gave the same response as above.)

Reviewer 3 Report

In this review Iijima and colleagues summarize current literature on ICH related pancreatitis and cholangitis focusing on the role of endoscopic diagnosis.

The manuscript is well organized and well presented. English language is correct and readable. I would only like to mention that lines 78 and 81 are misleading and I would suggest to be revised.

The presentation is simple and acceptable. Figures are satisfactory, tables are necessary, but I would suggest a revision of Table 1 as in this form is confusing (better to contain all the results in one line for each case).

References are satisfactory.

A major issue I would like to point out, is that the title of the manuscript does not completely reflect the content. Title focuses on the role of endoscopy in the diagnosis of ICH pancreatitis and cholangitis, but the main text does not focus on that. The main text contains a well-organized and interesting revision of current bibliography referring to incidence, symptoms, diagnosis and treatment of these two ICH related complications without focusing especially on the role of endoscopy in diagnosis. Moreover, in the present form the manuscript does not come to any conclusion about the role or the indication of endoscopy in ICH related cholangitis and pancreatitis. Last but not least, if the manuscript focus on the role of endoscopy then information about treatment could be considered not necessary. For this reason I would suggest either a change of the title so as to reflect better the content or a change in the content of the manuscript so as to clearly focus on the role of endoscopy in the diagnosis of ICH related pancreatitis and cholangitis.

Author Response

Thank you very much for your constructive and helpful comments to improve the impact of our manuscript. According to your comments and suggestions, we have modified our manuscript, please see the attachment.

Round 2

Reviewer 3 Report

The revised version of the manuscript is coherent and well presented. The manuscript has considerably benefited from the change in the title which now better and clearly reflects the content.